# INTERVENTION-BASED CAUSAL DISCRIMINATION DISCOVERY AND REMOVAL

## ABSTRACT

Causal inference is a recent and widely adopted paradigm to deal with algorithmic discrimination. Building on Pearl's structure causal model, several causality-based fairness notions have been developed, which estimates the unfair causal effects from the sensitive attribute to the outcomes by incorporating the intervention or counterfactual operators. Among them, interventional fairness (i.e., $K$-Fair) stands out as the most fundamental and broadly applicable concept that is computable from observational data. However, existing interventional fairness notions fail to accurately evaluate causal fairness, due to their following inherent limitations: (i) the causal effects evaluated by interventional fairness cannot be uniquely computed; (ii) the violation of interventional fairness being zero is not a sufficient condition for a causally fair model. To address these issues, we firstly propose a novel causality-based fairness notion called post-Intervention Cumulative Ratio Disparity (ICRD) to assess causal fairness of the decision models. Subsequently, we present a fairness framework (`ICCFL`) based on the proposed ICRD metric. `ICCFL` firstly generates interventional samples, and then computes the differentiable approximation of the ICRD to train a causally fair model. Both theoretical and empirical results demonstrate that the proposed ICRD effectively assesses causal fairness, and `ICCFL` can better balance accuracy and fairness.

## 1 INTRODUCTION

Recent years have witnessed wide usage of decision models based on machine learning techniques across various high-stakes domains, such as loan approval Kozodoi et al. (2022), job hiring decision Faliagka et al. (2012), and healthcare Pfohl et al. (2019). However, the predictions made by these decision models have been highlighted to be prone to unfair towards certain individuals or sub-groups characterized by the sensitive attributes, e.g., race and age. To mitigate the discrimination of the decision models, various fairness-aware algorithms have been developed in response to different fairness criterions. Early fairness notions are mostly based on statistical correlations, which measure the statistical discrepancy between sub-groups or individuals determined by the sensitive attributes, such as demographic parity Dwork et al. (2012); Jiang et al. (2020), predictive parity Chouldechova (2017) and equalized odds Hardt et al. (2016). However, studies Kusner et al. (2017); Zuo et al. (2022) have clarified that statistical correlation-based fairness notions fail to distinguish between discriminatory and spurious correlations between the outcome and the sensitive attribute.

To address the limitations of correlation-based fairness notions, several fairness notions are defined from causality, which aim to measure the unfair causal effects of the sensitive attribute on decision, e.g., counterfacutal fairness Kusner et al. (2017), path-specific fairness Zhang et al. (2017; 2018), proxy fairness Kilbertus et al. (2017), and interventional fairness Salimi et al. (2019); Ling et al. (2024). Among them, interventional fairness is a fundamental and general concept that typically can be uniquely computed from observational data. It aims to measure the unfair effects of the sensitive attribute on decision along the paths specific by certain context. However, existing interventional fairness Salimi et al. (2019); Ling et al. (2024), canonically referred to $K$-*Fair* (KF), cannot accurately measure whether the decisions of a model are causally fair or not, due to its following limitations:

i) The value of $K$-Fair is sensitive to the decision threshold in the classification task, where the decision threshold is used by the model to classify their predictions as positive or negative based on the predicted probabilities. In practice, the choice of decision threshold often varies, which can lead to

fluctuations in $K$-Fair assessments and thus fail to accurately measure causal fairness of the model. ii) The value $K$-Fair being zero is not a sufficient condition for a model to be causally fair. As shown in Table 3 of our experiments, even though the value of $K$-Fair is low, there are noticeable differences in the predicted probability distributions across different sensitive groups.

To address the issues mentioned above, we propose a novel causal fairness notion called Intervention-based Cumulative Ratio Disparity (ICRD). Given any specific intervention on the context, ICRD measures the cumulative causal effects along prediction probabilities by intervening on the sensitive attribute. Our theoretical analysis show that our ICRD includes several desirable properties such that it can accurately measure the causal fairness of a model. Moreover, based on the proposed ICRD metric, we introduce an Intervention-based Cumulative Causality Fairness Learning approach (`ICCFL`). Specifically, `ICCFL` formalizes the objective function as a constrained optimization problem by incorporating the proposed ICRD metric into the prediction loss of the model. `ICCFL` firstly generates the interventional samples through the causal model. Subsequently, to train such a causally fair decision model, `ICCFL` uses a temperature-scaled Sigmoid function to provide a differentiable approximation of the intervention cumulative distribution function, and finally minimizes the cumulative distribution discrepancy intervened on the sensitive attribute and context. In this way, `ICCFL` can effectively approach causal fairness. The main contributions are listed as follows:

- We propose a novel causality-based fairness notion called ICRD to assess the post-interventional cumulative ratio disparity, which holds several desired theoretical properties and is more advantageous to existing intervention causal fairness notions.
- Based on the proposed ICRD metric, we introduce an intervention-based cumulative causal fairness approach (`ICCFL`) that generates causality guided interventional samples and approximates the intervention cumulative distribution to mitigate cumulative causal effects along prediction probabilities.
- Experiments on benchmark datasets show that `ICCFL` achieves better causal fairness than competitive fairness methods Grgic-Hlaca et al. (2016); Kusner et al. (2017); Wu et al. (2019); Grari et al. (2023), and the elimination of post-intervention cumulative ratio disparity is equivalent to achieving causal fairness.

## 2 RELATED WORK

**Fairness Notions.** Due to the widespread application of machine learning algorithms in high-risk domains, algorithmic fairness has garnered substantial attention Shui et al. (2022). Generally, fairness metrics can be divided into two main types: statistical fairness and causal fairness. Statistical fairness notions measure the independence between the sensitive attribute and decision Dwork et al. (2012), while causality-based fairness notions aim to assess the unfair causal effects of the sensitive attribute on decision. Compared to statistical notions, causal fairness concepts have gained considerable attention, owing to their capability to identify spurious correlations between variables and uncover the true effects of the sensitive attribute on decisions. For example, counterfactual fairness Kusner et al. (2017) investigates whether a model's decision changes when the sensitive attribute of an individual is altered to another value, while keeping all other variables unchanged. Path-specific fairness Zhang et al. (2018) aims to measure the unfair effects of the sensitive attribute on decision transmitted along certain paths. Although counterfactual fairness and path-specific fairness are nuanced metrics, they are susceptible to identifiability issues, meaning that causal effects cannot be uniquely determined from observational data. Furthermore, despite the testable of Interventional Fairness Salimi et al. (2019); Ling et al. (2024), which measures causal effects intervened on the sensitive attribute and context, it may fail to capture causal fairness in certain cases.

**Fair Machine Learning.** So far many methods have been proposed for various causality-based fairness notions. These causality-based approaches can be broadly categorized into pre-processing mechanism, in-processing mechanism and post-processing mechanism Su et al. (2022). *Pre-processing* mechanism aims to detect and mitigate the bias presented in data before training the models. For example, Jones et al. (2024) investigated the sources of dataset bias and showed how the causal nature of dataset has the impacts on the deep learning models. Finally, they proposed a three-step framework to infer the fairness in medical imaging. *In-processing* mechanism enforces the causality-based fairness constraint into the model training process to mitigate the unfair causal

effects. Garg et al. (2019) penalized the differences between the real-world samples and their corresponding counterfacutal samples through counterfacutal logit pairing. Grari et al. (2023) firstly leveraged the adversarial learning to infer counterfactuals, and then forced the counterfactual fairness into the prediction loss based on the augmentational data for achieving fairness. *Post-processing* mechanism updates the prediction of the decision model to mitigate the unfair effects. For instance, Mishler et al. (2021) post-processed the binary predictor to satisfy approximate counterfacutal equalized odds using doubly robust estimators. Despite these notable efforts on causality-based fairness, it is unclear whether these methods can improve causal fairness by reducing the cumulative causal effects along the prediction probabilities.

To response, we propose a post-intervention cumulative ratio disparity (ICRD) notion to capture such cumulative causal effects, and further introduce a fairness model `ICCFL` based on ICRD. Compared to existing methods, `ICCFL` offers an effective way to capture and mitigate the cumulative causal effect of sensitive attribute on the predictions. Through theoretical analysis and comparison with state-of-the-art (SOTA) methods, we show that the proposed ICRD establishes a strong connection to causal fairness. In addition, although SOTA methods perform well on existing causal fairness metrics, they still exhibit discriminatory behaviors. In contrast, our `ICCFL` achieves consistent results, effectively approaching causal fairness.

## 3 PRELIMINARIES

We use boldface uppercase $\mathbf{X}$ to describe a subset of attributes, lowercase $\mathbf{x}$ to denote the values assigned to a subset of attributes. Let $\mathcal{D} = \{V_i = (S_i)|1 \le i \le n\}$ be a dataset with $n$ individual data points. Without loss of generality, we represent $S = \{s^+, s^-\}$ as the sensitive attribute, where $s^+$ and $s^-$ are the advantaged and disadvantaged groups, respectively. $Y$ represents the binary decision attribute, and $\mathbf{X}$ represents the set of non-sensitive attributes. We assume $\tilde{y} \in [0, 1]$ is the predictive probability of the decision model $f : \mathbb{R}^d \to [0, 1]$ with the model parameter $\theta$.

### 3.1 CORRELATION-BASED FAIRNESS NOTIONS

Correlation-based fairness notions aim to capture the statistical differences in the behavior of decision models across different sensitive groups. For example, *Demographic Parity* Jiang et al. (2020) requires the predictions of the model are independent of the sensitive attribute. *Equalized Odds* Hardt et al. (2016) measures the differences in false positive rate and false negative rate between advantaged group and disadvantaged group. Other popular statistical fairness notions include *Predictive Parity*, *Conditional Statistical Parity*, etc Chouldechova (2017). Despite the development of correlation-based fairness notions, they are unable to distinguish between causal relationships and spurious correlations among variables. To address these challenges, some causality-based fairness notions have been proposed, which can capture the causal relationships between variables and the outcome with the underlying causal model, as discussed below.

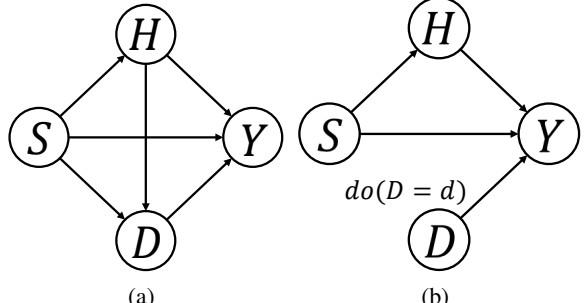

Figure 1: (a) is the ground truth causal graph; and (b) is the causal graph after performing intervention on $D$.

### 3.2 CAUSALITY-BASED FAIRNESS NOTIONS

**Causal Model.** Before discussing the causality-based fairness notions, we first introduce the causal model, which can be formally expressed as a quadruple $\mathcal{M} = \langle \mathbf{V}, \mathbf{U}, P(\mathbf{U}), \mathbf{F} \rangle$, where $\mathbf{V}$ is the set of observed variables, $\mathbf{U}$ is the set of unobserved exogenous variables, $P(\mathbf{U})$ is the probability distribution over $\mathbf{U}$, and $\mathbf{F}$ is the set of causal structure function $\mathbf{F} : \mathbf{U} \times \mathbf{V} \to \mathbf{V}$. A causal model is associated with a causal graph $G$, which describes the causally functional interactions between variables. There is an edge from $V_i$ to $V_j$, i.e., $V_i \to V_j$, iff $V_i$ causes $V_j$. As such, the joint

probability distribution of the set of observed variables can be decomposed as follows:

$$P(\mathbf{V}) = \prod_{V_i \in \mathbf{V}} P(V_i | Pa(V_i)) \tag{1}$$

where $Pa(V_i)$ are the parents of $V_i$ that directly cause $V_i$.

**d-separation and Faithfulness.** d-separation is a sufficient criterion that determines whether $\mathbf{M} \perp \mathbf{N}|_d\mathbf{Z}$, i.e., the observed variables $\mathbf{M}$ are independent of $\mathbf{N}$ conditioned on $\mathbf{Z}$. This is represented as all the paths between $\mathbf{M}$ and $\mathbf{N}$ being blocked by $\mathbf{Z}$ in the causal graph. *Faithfulness* is a fundamental assumption in the causal inference, ensuring that all observed conditional probability distribution $P$ of the dataset are reflected as d-separation in the corresponding causal graph, i.e., $\mathbf{M} \perp \mathbf{N}|_P\mathbf{Z}$ implies $\mathbf{M} \perp \mathbf{N}|_d\mathbf{Z}$; conversely, if $\mathbf{M} \perp \mathbf{N}|_d\mathbf{Z}$ implies $\mathbf{M} \perp \mathbf{N}|_P\mathbf{Z}$, the conditional probability distribution of the dataset and the causal graph are *Markov compatible*.

**Intervention.** An intervention on $V_i \in \mathbf{V}$, denoted by $do(V_i = v_i)$, means to break the causal function of variable $V_i$, and force $V_i$ to take a certain value $v_i$. Accordingly, all edges pointing to $V_i$ are discarded in the causal graph. We denote $P(y|do(V_i = v_i))$ as the post-intervention of $Y$ intervened by $do(V_i = v_i)$, which reflects the causal effects of $do(V_i = v_i)$. Specifically, given an intervention $do(S = s)$, the post-intervention distributions of an attribute $Y$ can be expressed as follows:

$$P(y|do(S = s)) = \sum_{\mathbf{z} \in \mathbf{pa}(s)} P(y|s, \mathbf{z}) \prod_{v \in \mathbf{V}'} P(v|\mathbf{pa}(v))\delta_{S=s} \tag{2}$$

where $\mathbf{z}$ is the parent of the intervention variable $S$, $\mathbf{V}' = \mathbf{V} \backslash \{S, Y\}$, and $\delta_{S=s}$ represents for any term involved $S$, the value of $S$ is taken as $s$. Note that if $\mathbf{Pa}(S) = \varnothing$, the post-intervention distribution is the same as conditional distribution, i.e., $P(y|do(S = s)) = P(y|S = s)$.

**Example 1.** *Consider the example mentioned in the introduction, which examines whether the admission decisions of the school exhibit discrimination towards gender. The corresponding causal graph is shown in Figure 1(a), where $S$ represents gender, $D$ represents the department, $H$ represents hobbies of individuals, and $Y$ stands for the admission decision. $S \rightarrow D$ indicates that applicants of different genders tend to apply to different departments (as evidenced by varying gender ratios across departments). Additionally, personal hobbies affect applicants' choice of department, and thus, there exists an edge $H \rightarrow D$. $H \rightarrow Y$ signifies that admission decisions take personal hobbies into account. As such, the joint probability distribution of the observed variables $\mathbf{V}$ can be expressed as follows:*

$$P(y, s, d, h) = P(y|s, d, h)P(d|s, h)P(h|s)P(s) \tag{3}$$

*When one performs intervention on $D$, i.e., forcing $D$ to take as $d$, according to Eq. equation 2 (as shown in Figure 1(b)), the post-intervention distribution of admission decision $Y$ can be expressed as follows:*

$$P(y|do(D = d)) = \sum_{s,h} P(y|D = d, s, h)P(h|s)P(s) \tag{4}$$

**Causal Fairness Notions.** With the intervention-operator, causality-based fairness notions aim to measure the causal effects of the sensitive attribute $S$ on the outcome $Y$ by intervening on $S$, e.g., counterfactual fairness (CF) Kusner et al. (2017), path-specific fairness (PSF) Zhang et al. (2018), and $K$-fairness (KF) Ling et al. (2024).

**Definition 1** (Counterfactual Fairness). *A decision model is considered counterfactual fairness if the prediction of the model for an individual remains unchanged when the sensitive attribute of such individual is altered to a different value (keeping the context, denoted by $\mathbf{O} = \mathbf{o}$, unchanged).*

$$P(\hat{y}|do(S = s^+), \mathbf{O} = \mathbf{o}) = P(\hat{y}|do(S = s^-), \mathbf{O} = \mathbf{o}) \tag{5}$$

**Definition 2** (Path-specific Fairness). *A decision model is considered path-specific fairness if the decision model removes the causal effects of the change of the sensitive attribute $S$ from $s^+$ to $s^-$ on the outcome $\hat{y}$ along the unfair paths $\pi$.*

$$P(\hat{y}|do(S = s^+|\pi, S = s^-|\bar{\pi})) = P(\hat{y}|do(S = s^-)) \tag{6}$$

*where $\pi$ is the set of unfair causal paths, the left-hand side of Eq. equation 6 represents the probability of the prediction after intervening on $S = s^+$ along the unfair path $\pi$, while intervening on $S = s^-$ along the remain paths $\bar{\pi}$.*

Table 1: The conditional probabilities under different decision thresholds.

| | $P(\hat{Y} = 1\|D =' A', S, H)$ | $S$ values | $H$ values | $P(H\|S)$ |
|---|---|---|---|---|
| | 0.2 | 1 | 1 | 0.2 |
| $\alpha = 0.5$ | 0.2 | 1 | 0 | 0.8 |
| | 0.2 | 0 | 1 | 0.8 |
| | 0.2 | 0 | 0 | 0.8 |
| | 0.06 | 1 | 1 | 0.2 |
| $\alpha = 0.6$ | 0.16 | 1 | 0 | 0.8 |
| | 0.07 | 0 | 1 | 0.8 |
| | 0.02 | 0 | 0 | 0.2 |

However, counterfacutal fairness and path-specific fairness may encounter identifiability issues, where the causal effects cannot be uniquely inferred from observational data. As a result, in this paper, we focus on *intervention-based causal fairness* notions, which can be testable from observational data. $K$-Fair (KF) is an exemplar intervention-based fairness notion.

**Definition 3** ($K$-fair). *Given a set of observed variables $\mathbf{K} \subseteq \mathbf{V}\backslash\{S,Y\}$, a decision model is considered $K$-fair if the predictions of the model are causally independent of the sensitive attribute conditioned on any context $\mathbf{K} = \mathbf{k}$.*

$$P(\hat{y}|do(S = s^+), do(\mathbf{K} = \mathbf{k})) = P(\hat{y}|do(S = s^-), do(\mathbf{K} = \mathbf{k})) \tag{7}$$

Although $K$-fair is a strong causality-based fairness notion that can be computable from observational data, it is insufficient for assessing the violation scores in term of causal fairness. Below, we discuss the limitations of existing interventional fairness notion, and subsequently, introduce our proposed fairness notion.

## 4 THE PROPOSED FAIRNESS NOTION AND METHOD

### 4.1 LIMITATIONS OF PREVIOUS NOTIONS

Exclusively leveraging existing intervention-based fairness notions (i.e., $K$-Fair) can result in unfair model, since a lower value of $K$-Fair may not accurately capture the true 'fairness' in decision-making.

**Limitation 1: Impacts of decision threshold.** *Threshold Rules* Corbett-Davies et al. (2023) are commonly applied in the decision process of the models for classification tasks. Specifically, for the classification task with binary classes, the decision models firstly produce predicted probabilities $\tilde{y}$ and then perform binary classification $\hat{y}$ based on the predefined decision threshold $\alpha$, i.e., $\mathbb{I}[\hat{y} \geq \alpha]$, where $\mathbb{I}[x]$ is an indicator function where $\mathbb{I}[x] = 1$ if $x$ is the true and $\mathbb{I}[x] = 0$ otherwise. For example, as for $\alpha = 0.5$, the decision is to admit the applicant if $\tilde{y} \geq 0.5$; conversely, the decision is to reject the applicant if $\tilde{y} < 0.5$. It is easy to show that the changes of the decision threshold can lead to variations in the measurement of $K$-Fair, as the predictions of the model depend on such threshold. Consequently, the assessment of $K$-Fair can be sensitive to the predefined decision threshold.

Let us reconsider Example 1, whose causal graph is shown in Figure 1. Without loss of generality, we assume that all variables are binary, where $S = 0$ denotes female and $S = 1$ means male. $Y = 1$ indicates the applicant is admitted, while $Y = 0$ indicates rejection. For concreteness, we consider the conditional probabilities shown in Table 1.

If one sets the decision threshold $\alpha = 0.5$ for classification, by performing intervention on $S = 0$ and $D =' A'$, the post-intervention distribution of $\hat{Y}$ can be computed as follows:

$$
\begin{aligned}
&P(\hat{Y} = 1|do(S = 0), do(D =' A')) \\
&= \sum_{h \in \{0,1\}} P(\hat{Y} = 1|S = 0, D =' A', H = h)P(H = h) \\
&= 0.2 \times 0.2 + 0.2 \times 0.8 = 0.2
\end{aligned}
\tag{8}
$$

Similarly, $P(\hat{Y} = 1 | do(S = 1), do(D =' A')) = 0.2$. Thus, the violation score of $K$-Fair is zero, indicating that the admission predictions are causally fair across different gender groups. However, when the decision threshold is set to 0.6, the admission predictions exhibit gender bias, as the violation score of $K$-Fair is 0.11. Consequently, when the decision threshold changes, $K$-Fair fails to accurately assess the model's fairness.

**Limitation 2: Insufficiency.** $KF = 0$ is only a necessary but insufficient condition for causal independence between the sensitive attribute and the outcome, conditioned on the given context $\mathbf{K} = \mathbf{k}$. That is, the causal effect of zero as evaluated by $K$-Fair requires that the predictions $\hat{y}$ are causally independence of the sensitive attribute given the context $\mathbf{K} = \mathbf{k}$. The reason is that the probability theory provided by Bisgaard & Sasvári (2000) demonstrating the identical probability functions are equivalent to having the same $r$-th moment for any $r$. However, $K$-Fair metric relies solely on the 1-th moment to measure the causal effects. Although $K$-Fair dose not detect any discrimination ($KF = 0$), the post-intervention distributions of the predictions follow different distributions. As a result, the decision model may still exhibit discrimination against the sensitive groups, even if the value of $K$-Fair is zero.

## 4.2 THE PROPOSED FAIRNESS NOTION

To address the limitations of existing intervention fairness notions, we propose a novel causality-based fairness notion called **Intervention-based Cumulative Rate Disparity** (ICRD for short). Specifically, ICRD aims to measure the cumulative causal effect of the sensitive groups on the model predictions, its formal definition is as follows:

**Definition 4** (ICRD). *Given a set of contexts $\mathbf{C}$, a decision model is considered as causality fairness if the following equation hold:*

$$\text{ICRD}(f) = \int_0^1 |F(\tilde{y}|do(S = s^+), do(\mathbf{C} = \mathbf{c})) - F(\tilde{y}|do(S = s^-), do(\mathbf{C} = \mathbf{c}))| \mathrm{d}\tilde{y} = 0 \quad (9)$$

*where $\tilde{y}$ is the prediction probabilities of the model, $F(\tilde{y}|do(S = s^+), do(\mathbf{C} = \mathbf{c}))$ represents the cumulative distribution function of the model prediction intervened by the sensitive attribute $do(S = s^+)$ and context $do(\mathbf{C} = \mathbf{c})$.*

$$F(\tilde{y}|do(S = s^+), do(\mathbf{C} = \mathbf{c})) = P(y \leq \tilde{y}|do(S = s^+), do(\mathbf{C} = \mathbf{c})) \quad (10)$$

*where $\tilde{y} \in [0, 1]$.*

Compared to existing interventional fairness notions, our fairness notion ICRD can more accurately capture the causal fairness of the decision models due to its several advantageous properties.

**Theorem 1.** *The fairness notion IRCD has the following properties:*
Property 1: *ICRD $= 0$ if and only if the model predictions $\hat{y}$ are causally independent of the sensitive variable $S$ conditioned on any given context $\mathbf{C} = \mathbf{c}$.*
Property 2: *The range of ICRD is within [0,1].*
Property 3: *ICRD is a continuous function.*

The proof of Theorem 1 is provided in the Appendix.

**Discussion.** Compared to $K$-Fair, ICRD satisfies the sufficiency condition for evaluating causal fairness. In addition, $K$-Fair measures the causal effects of the sensitive attribute on positive/negative model prediction with respect to decision threshold $\alpha$, which can be rewritten as $|F(\tilde{y}|do(S = s^+), do(\mathbf{C} = \mathbf{c})) - F(\tilde{y}|do(S = s^-), do(\mathbf{C} = \mathbf{c}))|$ with $y_0 = \alpha$. Therefore, ICRD encompasses $K$-Fair, and is the cumulative causal effect of $K$-Fair across all decision thresholds.

## 4.3 THE PROPOSED FAIRNESS METHOD

Based on the analysis mentioned above, we propose a novel fairness method called `ICCFL`, which learns a decision model $f_\theta$ with the parameters $\theta$ to mitigate the cumulative causal effects of the sensitive attribute on predictions for achieving causal fairness. To cope with it, `ICCFL` incorporates ICRD metric as the fairness constraint in the prediction loss. Formally, given a specific intervention on the context $do(\mathbf{O} = \mathbf{o})$, the optimization function of `ICCFL` can be expressed as follows:

$$\min_\theta \frac{1}{n} \sum_{i=1}^{n} \ell(\tilde{y}^i, y^i) + \lambda |\text{ICRD}(\tilde{y})| \quad (11)$$

---

**Algorithm 1** `ICCFL`: Intervention-based Cumulative Causal Fairness Learning

---

**Input**: The training data $\mathcal{D} = \{(s^i, \mathbf{x}^i, y^i) | 1 \leq i \leq n\}$, Causal Model $\mathcal{M}$, hyper-parameters $\lambda$ and $\tau$, learning rate $\eta$.
**Output**: Model parameters $\theta^*$
1: Sample $u$ from the distribution $P(U|S = s, \mathbf{X} = \mathbf{x})$
2: Generate interventional samples based on the inferred $u$ and causal model $\mathcal{M}$
3: **for** epoch $t = 1, 2, \cdots, T$ **do**
4:     **for** each mini-batch $\mathcal{B} \subseteq \mathcal{D}$ **do**
5:         Compute $\nabla_\theta \mathcal{L} = \nabla_\theta(\frac{1}{|\mathcal{B}|} \sum_{i=1}^{|\mathcal{B}|} \ell(\tilde{y}, \tilde{y}^i) + \lambda \widehat{\mathrm{ICRD}})$
6:         $\theta_{t+1} \leftarrow \theta_t - \eta \nabla_\theta \mathcal{L}$
7:     **end for**
8: **end for**
9: **return** model parameters $\theta^*$

---

where the key to optimizing this objective lies in assessing the cumulative post-intervention distribution $F(\tilde{y}|do(S = s), do(\mathbf{C} = \mathbf{c}))$ in $\mathrm{ICRD}(\tilde{y})$. To achieve this goal, `ICCFL` can utilize the Causal VAE Joo & Kärkkäinen (2020) to infer the distribution of exogenous variables $P_\mathcal{M}(U|S = s, \mathbf{X} = \mathbf{x})$, and then leverages such distribution and causal model $\mathcal{M}$ to generate the interventional samples with the interventions $(do(S = s^+), do(\mathbf{C} = \mathbf{c}))$ and $(do(S = s^-), do(\mathbf{C} = \mathbf{c}))$. Without loss of generality, we assume $\{\tilde{y}_+^1, \cdots, \tilde{y}_+^{n_+}\}$ with $n_+$ data points are the prediction probabilities for the sample under intervention $(do(S = s^+), do(\mathbf{C} = \mathbf{c}))$, while $\{\tilde{y}_-^1, \cdots, \tilde{y}_-^{n_-}\}$ with $n_-$ data points are the prediction probabilities for the sample under intervention $(do(S = s^-), do(\mathbf{C} = \mathbf{c}))$.

Subsequently, `ICCFL` can evaluate the term $\mathrm{ICRD}(\tilde{y})$ in Eq. equation 11 as follows:

$$\mathrm{ICRD}(\tilde{y}) = |\frac{1}{n_+} \sum_{i=1}^{n_+} \mathbb{I}(\tilde{y}_+^i \leq \tilde{y}) - \frac{1}{n_-} \sum_{i=1}^{n_-} \mathbb{I}(\tilde{y}_-^i \leq \tilde{y})| \tag{12}$$

where $\mathbb{I}(x)$ is the indicator function.

However, Eq. equation 12 is not differentiable with respect to the model parameters, resulting in optimization difficulties. To solve this problem, we perform a differentiable approximation mapping on the Eq. equation 12.

$$\widehat{\mathrm{ICRD}}(\tilde{y}) = |\frac{1}{n_+} \sum_{i=1}^{n_+} \sigma_\tau(\tilde{y} - \tilde{y}_+^i) - \frac{1}{n_-} \sum_{i=1}^{n_-} \sigma_\tau(\tilde{y} - \tilde{y}_-^i)| \tag{13}$$

where $\sigma_\tau(x) = \frac{1}{1+\exp(-\tau x)}$ is the mapping function, and $\tau$ is the hyper-parameter. Notably, when $\tau$ tends to infinity, the $\widehat{\mathrm{ICRD}}(\tilde{y})$ converges to the $\mathrm{ICRD}(\tilde{y})$ as follows.

**Theorem 2.** *As $\tau \to \infty$, $\widehat{\mathrm{ICRD}}(\tilde{y}) \to \mathrm{ICRD}(\tilde{y})$.*

The proof of Theorem 2 is given in the Appendix.

As a result, `ICCFL` can train a causally fair model by replacing $\mathrm{ICRD}(\tilde{y})$ with $\widehat{\mathrm{ICRD}}(\tilde{y})$ in Eq. equation 13. The overall procedure of `ICCFL` is presented in Algorithm 1. Lines 1-2 generate interventional samples based on causal model $\mathcal{M}$. Subsequently, at each epoch $t$, Line 5 computes the gradients of the model parameters for each sample with a mini-batch, and Line 6 updates the model parameters to reduce unfair cumulative effects caused by the sensitive attribute.

## 5 EXPERIMENTS

### 5.1 EXPERIMENTAL SETUP

In this section, we conduct experiments to evaluate the effectiveness of our `ICCFL` using real-world datasets (Adult, Dutch and Law School) Asuncion et al. (2007). The Adult dataset consists of 48,842 samples with 11 variables, where we treat '*sex*' as the sensitive attribute, 'education' as the context

Table 2: Accuracy and fairness results of our proposed ICCFL and the compared methods on real-world datasets. ○/● indicates that ICCFL is statistically worse/better than the compared method by student pairwise $t$-test at 95% confidence level. The best results are highlighted with **bold**, and the sub-optimal results are highlighted with underline.

| | Adult | | | Dutch | | | Law School | | |
|---|---|---|---|---|---|---|---|---|---|
| | Acc.↑ | $K$-Fair ↓ | ICRD ↓ | Acc.↑ | $K$-Fair ↓ | ICRD ↓ | MAE↓ | $K$-Fair ↓ | ICRD ↓ |
| Baseline | 0.766○ | 0.204● | 0.326● | 0.784○ | 0.198● | 0.232● | 0.734 | 0.344● | 0.397● |
| Unaware | 0.765○ | 0.167● | 0.303● | 0.776○ | 0.187● | 0.238● | 0.746 | 0.186● | 0.225● |
| A3 | 0.736 | 0.134● | 0.263● | 0.757 | 0.166● | 0.234● | 0.758 | 0.158● | 0.176● |
| CFB | 0.747 | **0.051** | 0.166● | 0.768 | 0.047 | 0.139● | 0.752 | **0.031** | 0.094 |
| ALCF | 0.751 | 0.076 | 0.174 | 0.772○ | 0.038 | 0.144● | 0.748 | 0.037 | 0.103● |
| ICCFL | 0.742 | 0.067 | **0.061** | 0.760 | **0.016** | **0.022** | 0.753 | 0.044 | **0.027** |

variable and '*income*' as the decision variable. We consider the causal graph introduced by Wu et al. (2019) as the ground truth, which is shown in Figure 1(a). The Dutch dataset contains 60,421 samples with 12 variables, where we also treat '*sex*' as the sensitive attribute, 'country_birth' as the context variable and '*occupation*' as the decision variable. The corresponding ground truth causal graph is given by Zhang et al. (2018) (shown in Figure 1(b)). The Law school dataset consists of 20,412 records, where we treat 'race' as the sensitive attribute, 'entrance exam socres' as the context variable, and 'first-year average grade' as the decision variable. We consider the causal graph introduced by Kusner et al. (2017) (level-2 causal model) as the ground truth. We use *Accuracy* (for classification tasks) and *mean absolute error (MAE)* (for regression tasks) as the metrics to measure the prediction performance of the models, and $K$-Fair (KF) and ICRD as the metrics to assess fairness.

The experiments are conducted by comparing ICCFL against:

- Baseline, which use all variables to train the model without fairness constraints;
- Unaware Grgic-Hlaca et al. (2016), uses the variables except the sensitive attribute to train the model;
- A3 Kusner et al. (2017), assumes the causal model as the additive noise model, and assesses the noise term, which is then used to train the model;
- CFB Wu et al. (2019), incorporates interventional fairness into the training process;
- ALCF Grari et al. (2023), employs adversarial learning with a causal model to achieve causal fairness.

All compared methods use the same ReLU neural network with four hidden layers as the base model, and thus, they have the same number of model parameters.

For all used datasets, we split the dataset into training, validation, and test sets with proportions of 70%, 10%, and 20%, respectively. We report the average results and standard deviations over ten run times of the experiments. As for the selection of the hyper-parameters for all compared methods, we use the grid search strategy (ranges specified in Table A1) on the validation set to find the best hyper-parameters. In this paper, we use Pyro Bingham et al. (2019) to construct the causal models of Adult, Dutch and Law School datasets.

## 5.2 PERFORMANCE COMPARISON

In this section, we study the trade-off between accuracy and fairness of the above methods. Table 2 presents the performance in term of accuracy and fairness of each method. From these results, we can observe that:

i) ICCFL outperforms compared methods in term of fairness, and achieves a higher (or similar) accuracy than comparisons. This indicates that our ICCFL can effectively mitigate the bias cumulative

Table 3: Accuracy and fairness results of ICCFL and its variant on real-world datasets. ○/● indicates that ICCFL is statistically worse/better than the compared method by student pairwise $t$-test at 95% confidence level. The best results of fairness are highlighted with **bold**.

| | Adult | | | | Dutch | | | |
|---|---|---|---|---|---|---|---|---|
| | Acc.↑ | $K$-Fair↓ | ICRD↓ | MMD↓ | Acc.↑ | $K$-Fair↓ | ICRD↓ | MMD↓ |
| ICCFL-KF | 0.744 | **0.038**○ | 0.133● | 12.748● | 0.763 | 0.018 | 0.117● | 15.774● |
| ICCFL | 0.742 | 0.067 | **0.061** | **6.634** | 0.760 | **0.016** | **0.022** | **5.132** |

causal effects of the predictions to improve the causal fairness.

ii) Compared to Baseline, CFB and ALCF exhibit a reduction in fairness violation, and achieve an acceptable balance between fairness and accuracy. This suggests that utilizing traditional interventional fairness helps to reduce unfair cumulative causal effects of the model predictions. However, compared to ICCFL, their lower performance in term of ICRD and $K$-Fair highlights the limitations of these approaches in achieving causal fairness. In addition, among fairness-aware methods, A3 exhibits the worst trade-off between accuracy and fairness, which shows that unrealistic causal model assumptions can mislead the training of fair classifier.

iii) Although Baseline achieves the highest accuracy performance, it performs the poorest in fairness. This is because the primary objective of Baseline is to optimize accuracy. In addition, Unaware mitigates discrimination by excluding the sensitive attribute, it still struggles to reduce unfair effects caused by descendants of the sensitive attribute. In contrast, ICCFL can mitigate the negative impacts of the sensitive attribute and its descendants by minimizing cumulative causal disparity.

## 5.3 THE BENEFIT OF ICRD

To further study the effectiveness of the proposed ICRD metric for assessing the causal fairness, we consider an variant of ICCFL, denoted by ICCFL-KF, which takes $K$-Fair into accounts during the model training. Recall that a decision model is causally fair if there is no disparity in the distribution of prediction probabilities on different interventional samples generated by the ground truth causal model. To this end, we leverage Maximum Mean Dis-

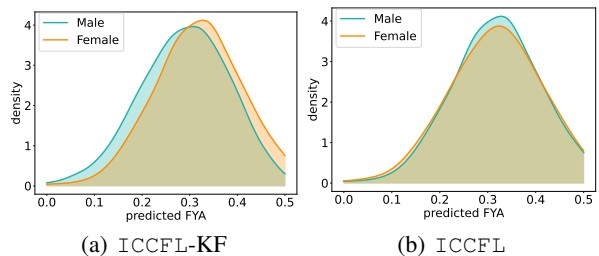

(a) ICCFL-KF        (b) ICCFL

Figure 2: Density distribution of predicted FYA for ICCFL-KF and ICCFL.

crepany (MMD) to measure such distribution divergence, where MMD first applies the kernel embedding techniques to map the samples into a Reproducing Kernel Hilbert Space, and subsequently, uses the Gaussian kernel to compare the samples. The results are presented in Table 3. We also show the probability density function of predicted First-Year Average grade (FYA) in Law School dataset, under both ICCFL and ICCFL-KF, in Figure 2, where the blue curve represents the predictions for samples of Male group and orange curve represents the predictions for samples of Female group. We have the following conclusions:

i) ICCFL obtains clearly better ICRD results across real-world datasets, and also achieves better or comparable performance in term of $K$-Fair. This suggests that minimizing cumulative causal disparity along predictions improves $K$-Fair. Such observation aligns with the properties of ICRD, i.e., ICRD metric generalizes $K$-Fair.

ii) Although ICCFL-KF obtains a small violation score of $K$-Fair, it exhibits significant differences in predictions (a large MMD) across different sensitive groups. The results confirm that a small violation of $K$-Fair may not represent high-level causal fairness. In other words, $K$-Fair is not a sufficient condition for causal fairness.

iii) We preliminary observe that ICRD and MMD exhibit similar patterns of variation, with lower ICRD values aligning with smaller MMD values presented in ICCFL. In addition, ICCFL maintains the model's behavior consistently across different sensitive groups. To further confirm this observa-

tion, in the next section, we conduct hyper-parameter analysis experiments by varying the value of $\lambda$ in Eq. (11). From the results shown in Figure 3, we can draw the similar conclusions.

## 5.4 HYPER-PARAMETER ANALYSIS

**Impacts of $\lambda$.** In our proposed ICCFL, $\lambda$ is a crucial hyper-parameter that controls the trade-off between the model performance in term of accuracy and fairness. As such, in this section, we conduct experiments on Adult dataset (similar patterns can be observed in Dutch dataset) to analyze the impact of hyper-parameter $\lambda$ by varying $\lambda$ within $\{0.05, 0.5, 2.0, 10, 30\}$. The results under different input values of $\lambda$ are shown in Figure 3. We can observe that:

i) As expected, when $\lambda$ increases, ICCFL places greater emphasis on model fairness. As a result, ICCFL achieves a better causal fairness at the expense of lower accuracy.
ii) ICRD and MMD exhibit similar trends, with a decrease in ICRD aligning with a reduction in MMD. This correlation suggests that as the ICRD value diminishes, the model's predictions become increasingly fair for sensitive groups, consistent with the property 3 outlined in Theorem 1. We can conclude that when the ICRD value reaches to zero, the decision model achieves causal fairness.

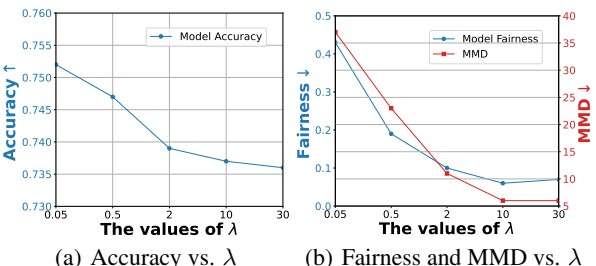

(a) Accuracy vs. $\lambda$      (b) Fairness and MMD vs. $\lambda$

Figure 3: Performance of ICCFL vs. hyper-parameter $\lambda$.

**Impacts of $\tau$.** The hype-parameter $\tau$ in our proposed ICCFL is also crucial to approximate the true post-intervention cumulative causal effects. To verify the impacts of this hype-parameter, we also conduct experiments on Adult dataset by varying $\tau$ within $\{3, 10, 20, 100\}$. The corresponding results are shown in Figure 4. We can observe that:
i) As $\tau$ increases, the evaluation errors of proposed metric ICRD decrease, in line with Theorem 2.
ii) When $\tau$ value is too large (e.g., $\tau = 100$), the gradient may vanishes, thereby restricting the model's learning capacity and hindering convergence during model updating.
iii) The moderate $\tau$ values (e.g., $\tau = 10$) are recommended to effectively balance the model performance and the gradient problem during optimization.

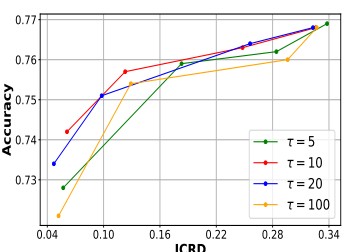

Figure 4: Accuracy and fairness trade-offs as $\tau$ varies. Each symbol represents the average results of ten runs at different values of $\lambda$.

## 6 CONCLUSION

In this paper, we delve into more effective metric for evaluating the causal fairness of a decision model through intervention techniques. We uncover the limitations of existing interventional fairness, particularly $K$-Fair, revealing that these fairness notions often fall short in capturing the unfair causal effects of sensitive attributes on outcomes. Specifically, we show that the value of $K$-Fair being zero does not sufficiently guarantee the causal fairness. Based on these observations, we introduce a novel intervention fairness notion (ICRD), which measures the post-intervention cumulative causal effects along the prediction probabilities for any intervention on the context $do(\mathbf{C} = \mathbf{c})$. Subsequently, we present a causality-based fairness framework to approximately assess and reduce ICRD values for achieving causal fairness. Experiments on real-world datasets confirm the effectiveness of our metric and framework.

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

## A    THE CAUSAL GRAPHS OF REAL-WORLD DATASETS

Figure 1(a) shows the ground truth causal graph of Adult dataset, and Figure 1(b) shows the ground truth causal graph of Dutch dataset.

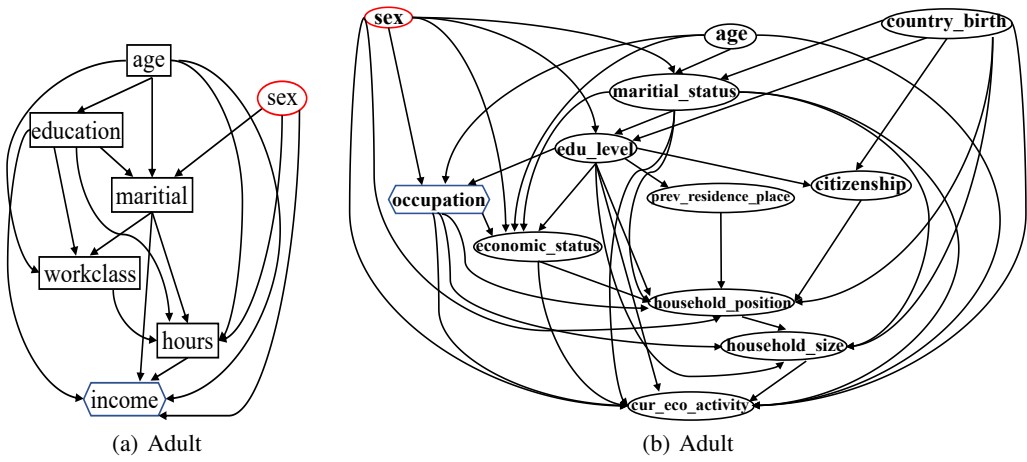

(a) Adult                               (b) Adult

Figure A1: The ground true causal models of Adult and Dutch.

## B    HYPER-PARAMETER SETTINGS

We use the grid search strategy on the validation set to find the best hyper-parameters for all compared methods. We verify all methods with their hyper-parameters as listed in Table A1.

Table A1: Method specific hyper-parameters: $lr$ is the learning rate of the corresponding model, $\tau$ is the fairness threshold (CFB), $\lambda$ is the parameter of the fairness constraint (ALCF).

| Method | Hyper-parameters |
|---:|---|
| BL | $lr \in \{0.001, 0.005, 0.01, 0.05, 0.1, 0.2, 0.5\}$ |
| Unaware | $lr \in \{0.001, 0.005, 0.01, 0.05, 0.1, 0.2, 0.5\}$ |
| A3 | $lr \in \{0.001, 0.005, 0.01, 0.05, 0.1, 0.2, 0.5\}$ |
| CFB | $lr \in \{0.001, 0.005, 0.01, 0.05, 0.1, 0.2, 0.5\}$, $\tau = 0.05$ |
| ALCF | $\lambda \in \{0.0, 0.2, 0.4, 0.6, 0.8\}$ |
| ICCFL | $\lambda \in \{0.05, 0.5, 1.0, 2, 5, 10, 20\}$, $\tau \in \{5, 10, 20, 30, 50\}$ |

## C    THE PROOF OF THEOREM 1

i) **The proof of Property 1:**
If the model predictions satisfy causal fairness, the predictive probabilities under different interventions on the sensitive attribute should be the same. That is to say, given any different interventions on the sensitive attribute, the post-intervention distributions of the predictive probability conform to the identical distribution, i.e., $\forall \tilde{y} \in [0, 1], F(\tilde{y}|do(S = s^+), do(\mathbf{C} = \mathbf{c})) = F(\tilde{y}|do(S = s^-), do(\mathbf{C} = \mathbf{c}))$. Then, according to Eq. equation 9 and Eq. equation 10, we can obtain $\text{ICRD}(\tilde{y}) = 0$.

Conversely, according to the Definition 4 for $\text{ICRD}(\tilde{y})$, the following holds:

$$\text{ICRD}(\tilde{y}) = 0 \Rightarrow F(\tilde{y}|do(S = s^+), do(\mathbf{C} = \mathbf{c})) = F(\tilde{y}|do(S = s^-), do(\mathbf{C} = \mathbf{c})), \forall \tilde{y} \in [0, 1] \quad \text{(A1)}$$

Therefore, ICRD is a sufficient and necessary condition for the causal fairness:

$$\text{ICRD}(\tilde{y}) = 0 \Leftrightarrow F(\tilde{y}|do(S = s^+), do(\mathbf{C} = \mathbf{c})) = F(\tilde{y}|do(S = s^-), do(\mathbf{C} = \mathbf{c})), \forall \tilde{y} \in [0, 1] \tag{A2}$$

ii) **The proof of Property 2:**
If the decision model is causal fairness, i.e., $\forall \tilde{y} \in [0, 1], F(\tilde{y}|do(S = s^+), do(\mathbf{C} = \mathbf{c})) = F(\tilde{y}|do(S = s^-), do(\mathbf{C} = \mathbf{c}))$, then $\text{ICRD}(\tilde{y}) = 0$. Besides, without loss of the generality, let $F_{s^+} = \arg\max F(\tilde{y}|do(S = s^+), do(\mathbf{C} = \mathbf{c})) = 1$ and $F_{s^-} = \arg\min F(\tilde{y}|do(S = s^-), do(\mathbf{C} = \mathbf{c})) = 0$, then we can obtain $\text{ICRD}(\tilde{y}) = 1$. Thus, we have $\text{ICRD}(\tilde{y}) \in [0, 1]$.

iii) **The proof of Property 3:**
It is easy to verify the continuity condition of ICRD, as the estimation of the cumulative distribution function is continuous with respect to the model predictions, and our proposed fairness metric $\widehat{\text{ICRD}}$ is also continuous with respect to the estimations of the cumulative distribution function.

# D  THE PROOF OF THEOREM 2

For any $\tilde{y} \in [0, 1]$, we can obtain

$$\lim_{\tau \to \infty} \sigma(\tilde{y} - \tilde{y}^i) = \frac{1}{1 + \exp(-\tau(\tilde{y} - \tilde{y}^i))} = \begin{cases} 1 & \text{if } \tilde{y}^i < \tilde{y}, \\ \frac{1}{2} & \text{if } \tilde{y}^i = \tilde{y}, \\ 0 & \text{if } \tilde{y}^i > \tilde{y}, \end{cases} \tag{A3}$$

Then under any intervention on the sensitive attribute and contexts $(do(S = s), do(\mathbf{C} = \mathbf{c}))$, we have

$$\lim_{\tau \to \infty} \sum_{i=1}^{n} \sigma_\tau(\tilde{y} - \tilde{y}_{S \leftarrow s}) = \sum_{i=1}^{n} \lim_{\tau \to \infty} \sigma_\tau(\tilde{y} - \tilde{y}_{S \leftarrow s}) = \sum_{i=1}^{n} \mathbb{I}(\tilde{y}_{S \leftarrow s} \leq \tilde{y}) \tag{A4}$$

According to Eq. equation A4, we can obtain

$$\lim_{\tau \to \infty} \widehat{\text{ICRD}}(\tilde{y}) = |\frac{1}{n_+} \sum_{i=1}^{n_+} \mathbb{I}(\tilde{y}_-^i \leq \tilde{y}) - \frac{1}{n_-} \sum_{i=1}^{n_-} \mathbb{I}(\tilde{y}_-^i \leq \tilde{y})| = \text{ICRD}(\tilde{y}) \tag{A5}$$