# OpenReview forum: "Intervention-based Causal Discrimination Discovery and Removal"
_ICLR.cc/2025/Conference — Submitted to ICLR 2025_

### Official Review · Reviewer_zFL6 · 2024-11-03

**Soundness:** 2
**Presentation:** 2
**Contribution:** 2
**Rating:** 3
**Confidence:** 4

**Summary:**

This paper demonstrates the limitations of the existing interventional fairness and then proposes a new causal fairness metric called Intervention-based Cumulative Rate Disparity (ICRD). ICRD aims to measure the post-intervention cumulative causal effects along the prediction probabilities for any intervention on the context. In addition to defining this metric, the authors propose an algorithm designed to achieve ICRD.

**Strengths:**

- It is reasonable and meaningful to uncover the limitations of the existing fairness notions and propose a new one.
- The experimental results show the effectiveness of the proposed method.

**Weaknesses:**

- The motivation behind ICRD is somewhat ambiguous. Specifically, regarding sufficiency, how is a condition defined as ‘sufficient’ for evaluating causal fairness? It seems that the sufficiency aspect depends significantly on the particular causal fairness definition in use, and the current explanation feels unclear on this point. The insufficiency aspect could benefit from greater elaboration.
- Lines 313–314 state that “ICRD encompasses K-Fair and represents the cumulative causal effect of K-Fair across all decision thresholds,” but this claim is difficult to interpret without additional clarification. Similarly, it is unclear how Table 1 was generated or how the decision threshold impacts outcomes. Could the authors further clarify these aspects?
- Finally, I am unconvinced that the decision threshold’s impact constitutes a limitation of K-Fair. K-Fair requires two distributions to be equivalent; hence, it is unclear how the decision threshold would influence this requirement. More discussion on this would be valuable to fully understand the claimed limitation.

**Questions:**

Please refer to the above weaknesses.

---

### Official Review · Reviewer_P7r5 · 2024-11-04

**Soundness:** 3
**Presentation:** 3
**Contribution:** 2
**Rating:** 5
**Confidence:** 3

**Summary:**

Motivated by shortcomings of existing interventional fairness notions, this paper proposed a new causality-based fairness notion called post-Intervention Cumulative Ratio Disparity (ICRD). ICRD measures the cumulative causal effects along prediction probabilities by intervening on the sensitive attribute. The authors explained ICRD’s superior properties over existing intervention causal fairness notions. Additionally, they developed a new fairness framework based on ICRD: Intervention-based Cumulative Causality Fairness Learning approach (ICCFL) formulates a constrained optimization problem where the ICRD metric is included in the prediction loss of the model. Empirical evidence from comparing ICCFL with several benchmark methods demonstrated that ICCFL could attain better causal fairness.

**Strengths:**

1. The paper improved the existing interventional fairness notion, K-Fairness, in a comprehensive way that both develops a new fairness notion and proposes an algorithm for applying the new fairness notion. The authors also provided relevant theoretical support for the validity of both ICRD and ICCFL, which add to the technical soundness of the paper.

2. The paper provided useful details in the experiment evaluation of the ICCFL method: Section 5.3 offered empirical evidence for the benefit of ICRD, and Section 5.4 discussed observations related to hyperparameter choice in ICCFL.

**Weaknesses:**

1. Given that the differences between the K-Fair notion and the new ICRD notion are somewhat subtle, the paper could benefit from clearer explanations. For example, Example 1 used to discuss limitation 1 might be applied again after introducing ICRD to illustrate how ICRD applies here, such as, what are the possible contexts C in this example. On a related note, although the ICRD notion has clear advantages over the K-Fair notion, it is unclear whether these advantages alone justify adding ICRD to the already large number of causal fairness definitions. It would be helpful to discuss the benefits of ICRD as a causal fairness definition in general.

2. The ICRD notion centers on disparity in the cumulative causal effects. This is not necessarily desirable for understanding discrimination, as we may be more interested in dissecting the causal effects associated with specific scenarios. It would be helpful to discuss potential insufficiencies of the ICRD notion, for example, when ICRD may not be identifiable, when enforcing ICRD to be 0 may be too restrictive for fairness.

**Questions:**

1. How does the ICRD notion address limitation 1?

2. Why does interventional fairness have fewer identifiability challenges compared to the counterfactual fairness and path-specific fairness, as mentioned on line 228, page 5?

3. Can the ICCFL method be compared with any benchmark methods using other causal fairness notions, such as, path-specific fairness? This might reveal interesting observations on the comparison between ICRD and other non-intervention based causal fairness measures.

---

### Official Review · Reviewer_GHKa · 2024-11-04

**Soundness:** 3
**Presentation:** 3
**Contribution:** 3
**Rating:** 6
**Confidence:** 2

**Summary:**

The paper introduces a new fairness metric called Intervention-based Cumulative Ratio Disparity (ICRD), which aims to address limitations in existing causal fairness metrics (K-Fair) by measuring cumulative causal effects along prediction probabilities by intervening on sensitive attributes. Additionally, the authors propose a fairness framework, ICCFL, which incorporates the ICRD metric to train fairer models. Through theoretical and empirical analyses, the paper demonstrates that ICCFL better balances fairness and accuracy than existing fairness-aware algorithms across multiple datasets.

**Strengths:**

1. The authors clearly illustrated the limitations in existing interventional fairness metrics, and the related works section is comprehensive and easy to follow.

2. The proposed formulation of ICRD is sound and the authors provide the theoretical analysis on how ICRD addresses the limitations of existing causal fairness metrics.

3. The authors proposed a fairness framework, ICCFL, which incorporates a differentiable approximation of the ICRD metric to enable efficient training.

**Weaknesses:**

1. The proposed method assumes the causal model is known, which may be a strict assumption. It would be great for the authors to discuss the sensitivity of the proposed metric and framework to potential causal graph misspecification.

2. This paper assumes the sensitive attribute is binary. Could the proposed metric be extended to handle multiple sensitive attributes?

3. The method leverages causal generative models to infer the distribution of exogenous variables. It would be useful to explore the robustness of the approach when estimating interventional distributions with different causal generative models.

**Questions:**

Please refer to the questions in Weaknesses part.

---

### Official Review · Reviewer_mDmt · 2024-11-05

**Soundness:** 3
**Presentation:** 1
**Contribution:** 2
**Rating:** 3
**Confidence:** 4

**Summary:**

This paper adds to the causal fairness literature by proposing a new metric to measure unfairness and a strategy for training fair models. It follows previous work by Salimi et al (2019) and Ling et al (2023) on interventional fairness or K-fairness. An algorithm is K-fair if interventions on the sensitive attribute do not change the predictions, while also causally conditioning on a given context K. The current paper extends this definition by applying a 1-Wasserstein distance to the difference between the interventional distributions, with interventions on the sensitive attribute. The proposed training strategy is empirical risk minimization with a penalty term added using the aforementioned 1-W. distance. The paper includes a few basic theoretical results and experiments comparing the method with several alternate methods on several datasets.

**Strengths:**

The overall setting is well-chosen and the contribution appears to be solid.

**Weaknesses:**

Compared to the existing work I believe this paper is somewhat incremental. The novelty is not high. The experiments are OK. The presentation and explanations of both current work and its context in related literature are not very clear.

**Questions:**

Since the main contribution of this paper is to build on the K-fair definition, what could you do specifically to include a more comprehensive and clear explanation of K-fair? How is the set of contexts C chosen, and which contexts were used in the experiments?

---

### Official Review · Reviewer_CYP9 · 2024-11-08

**Soundness:** 2
**Presentation:** 2
**Contribution:** 2
**Rating:** 5
**Confidence:** 3

**Summary:**

This paper proposes a novel causality-based fairness notion called post-intervention Cumulative Ratio Disparity (ICRD) to assess the causal fairness of the decision models, and then presents a causal framework based on ICRD. The theoretical and empirical results show that ICRD can assess causal fairness and the causal framework can better balance accuracy and fairness.

**Strengths:**

- The paper proposes a novel notion to measure causal fairness. This notion makes intuitive sense and seems easy to implement.
- The paper proposes a new algorithm to train a model, where the causal fairness notion is cast as a regularization term.
- On several empirical datasets, the proposed algorithm seems to perform best in terms of causal fairness, as compared to several benchmarks.

**Weaknesses:**

- It seems that the proposed algorithm is not very competitive as compared to benchmarks if one primarily cares about conventional metrics e.g., K-fair and accuracy.
- The theoretical results are quite intuitive, and the proof is straightforward. It would be helpful to the contributions of the paper, and why the contributions are nontrivial to obtain.
- The references of this paper do not contain a single ICLR paper. It would be helpful to better demonstrate the fit of this paper to ICLR.

**Questions:**

Please address the weaknesses above.

---

### Meta-Review · Area_Chair_9JDq · 2024-12-21

**Metareview:**

The paper proposed a fairness metric that measures cumulative causal effects along prediction probabilities by intervening on sensitive attributes

Strengths:

+ Studies an important limitations in existing interventional fairness measures

Weaknesses:

+ The contribution appears somewhat incremental, with reviewers questioning whether the advantages of ICRD justify adding another fairness definition to an already crowded field

+ The paper makes strong assumptions about knowing the causal model and having binary sensitive attributes, with limited discussion of robustness to model misspecification

+ The method underperforms on conventional metrics (accuracy and K-fairness) compared to benchmarks, and the paper lacks comparisons with non-intervention based causal fairness measures

**Additional Comments On Reviewer Discussion:**

The reviewers are largely in agreement that the paper in its current form does not meet the acceptance threshold.

---

### Decision · Program_Chairs · 2025-01-22

Reject